# The ISC Bulletin as a comprehensive source of earthquake source mechanisms

Konstantinos Lentas [1], Domenico Di Giacomo [1], James Harris [1], and Dmitry A. Storchak [1]

[1]International Seismological Centre, Pipers Lane, Thatcham, Berkshire, RG19 4NS, UK

**Correspondence:** Konstantinos Lentas (kostas@isc.ac.uk)

**Abstract.** In this article we summarize the availability of earthquake source mechanisms in the Bulletin of the International Seismological Centre (ISC). The bulletin in its current status contains ∼81,000 seismic events with only one associated mechanism solution, and ∼25,000 events with at least two associated source mechanisms. The main sources of earthquake mechanisms in the ISC Bulletin are reported solutions provided by data contributors, and ISC computed focal mechanisms based on first motion polarities. Given the importance of using pre-determined fault plane solutions in different types of studies, here we focus only on the reported mechanisms and we briefly discuss the methodologies adopted by major data providers to the ISC and investigate the intra-event variability of the source mechanisms. We conclude that the overall agreement among different earthquake mechanisms for the same event as reported by different sources can show a similarity coefficient as high as 80%, based on the rotation angles of their best fitting double couple solutions, for the majority of the cases. The earthquake source mechanisms discussed in this work are freely available within the ISC Bulletin websearch at http://doi.org/10.31905/D808B830.

## 1 Introduction

The International Seismological Centre (ISC, www.isc.ac.uk) currently collects station readings, hypocentre solutions and other earthquake bulletin data from approximately 150 agencies around the world. The ISC Bulletin contains over 7.6 million seismic events (mostly earthquakes, as well as chemical and nuclear explosions, mine blasts and mining induced events, and other types of seismic events), and approximately 256 million associated seismic station readings of arrival times, amplitudes, periods and first motion polarities (International Seismological Centre, 2018, database last accessed in March 2019).

Considerable effort is put into making sure that the station readings reported by different agencies belong to the correct seismic event. In the first instance, all parametric data sent to the ISC is collected and grouped automatically in unique seismic events. As soon as an event is created it is made openly available via the online ISC Bulletin (www.isc.ac.uk/iscbulletin). Secondly, the ISC analysts manually review (two/three years behind real-time) the collected station readings and hypocentre solutions for seismic events larger than approximately 3.5. If all conditions are met (details at http://www.isc.ac.uk/iscbulletin/review/), the ISC also recomputes location and magnitude (currently only MS and mb) by combining all the available phase arrival times and amplitude measurements, respectively. ISC location and magnitude procedures have recently been improved (Bondár and Storchak, 2011).

The ISC aims to increase the number of collected bulletins from national data centres or other sources (Willemann and Storchak, 2001) and improve its procedures in earthquake location and magnitude determinations (e.g., Bondár and Storchak, 2011; Di Giacomo and Storchak, 2015; Weston et al., 2018). As a result, the ISC Bulletin has proved to be a very useful resource for seismologists and geoscientists in general, as demonstrated by the vast use of ISC datasets in many research papers, including works on new tomographic models and global tectonics (e.g., Kennett et al., 1995; Rezapour and Pearce, 1998; Bormann et al., 2009; Hayes et al., 2012; Adam and Romanowicz, 2015; Zhan and Kanamori, 2016; Euler and Wysession, 2017; Lay et al., 2017). Recently the ISC has started to compute its own focal mechanisms (freely available in the reviewed bulletin) by using first motion polarities both from reported bulletins and picked automatically from waveform data (Lentas, 2017). In addition, the ISC Bulletin contains a substantial amount of source mechanisms (Fig. 1) calculated using different data and techniques as reported from various agencies working at local/regional and/or global scales, predominantly covering the period from mid 1970s till present.

In this paper, we aim to emphasize the availability of source mechanisms in the ISC Bulletin and discuss the different features of those solutions, aiming at helping ISC data users to decide how best to use the database according to the needs of their research.

## 2   Source mechanism contributions to the ISC Bulletin

There are currently 65 agencies in the ISC Bulletin which have reported in the past or continue to report source mechanism solutions to the ISC (Fig. 2). By using the term source mechanisms we refer to both moment tensor solutions (and their associated best fitting double couple mechanisms) and pure double couple mechanisms of a point source. Table 1 shows details of the type of reported source mechanism solutions by each agency.

Major contributors of global source mechanisms include the Global Centroid Moment Tensor Project (GCMT, www.globalcmt. org, Dziewonski et al., 1981; Ekström et al., 2012), the US National Earthquake Information Center (NEIC, or NEIS prior to 1984), and for regional earthquakes, the National Research Institute for Earth Science and Disaster Resilience (NIED) in Japan and the Pacific Northwest Seismic Network (PNSN).

Note that prior to data year 2006 the agency code HRVD (Harvard University) was used throughout the ISC Bulletin for GCMT solutions. Here we use a unique agency code for these source mechanisms and replace the HRVD agency code with the GCMT code throughout the ISC Bulletin. This is already done for the time period 1976-1979 covered by the first part of the ISC rebuild project (1964-1979, Storchak et al., 2017). After completion of the ISC rebuild project, all remaining HRVD source mechanism solutions will be available under the GCMT code. Moreover, moment tensor solutions for 76 intermediate depth earthquakes and 104 deep earthquakes from 1962 to 1976 have been added under the GCMT agency code (Chen et al., 2001; Huang et al., 1997).

Since the mid 1990s numerous other agencies, mainly national data centres, started reporting source mechanism solutions to the ISC. This has resulted in a steep increase of available mechanism solutions in the ISC Bulletin (Fig. 1). Nevertheless, the coverage and completeness of seismic events with associated source mechanisms is not uniform and primarily depends on

the tectonics and the associated seismicity in different regions, the station coverage and the practices of the reporting agencies (Fig. 3).

In May 2018 the ISC published in the online ISC Bulletin its first automatic focal mechanism solutions obtained from reported first motion polarities and automatic picks of waveform data. These are focal mechanisms for reviewed and relocated earthquakes in the Reviewed ISC Bulletin with $m_b^{ISC} \geq 4.5$, starting from data month January 2011. Since then new focal mechanism are routinely added for every data month added in the Reviewed ISC Bulletin. Moreover, we have published focal mechanism solutions obtained from reported polarities for the ISC relocated earthquakes ($m_b^{ISC} > 3.5$) covering the time period 1964 - 1984 as part of the rebuild project (Storchak et al., 2017) and focal mechanism solutions obtained from reported polarities in the ISS (1938-1963) Bulletins (see http://www.isc.ac.uk/projects/focalmechs/). The gap currently shown in Figure 2 between 1985 and 2010 is expected to be bridged gradually until the completion of the rebuild project.

Figure 4 shows the geographical and magnitude distribution of earthquakes with source mechanisms reported by the agencies which systematically send their mechanism solutions to the ISC (see also Fig. 2 for numbers of reported source mechanisms by agency and distribution in time). Local agencies are important to complement the results of global agencies as they cover events with lower magnitudes. For example, GCMT computes source mechanisms for global earthquakes with magnitude 5.0 and above, but also slighlty lower ($\sim 4.5$) depending on the area and station coverage. Similarly, NEIC covers earthquakes with a minimum magnitude of $M_W \sim 4.5$ on a global scale. IPGP reports earthquakes with magnitude $M_W$ 5.5 and above, whilst at the ISC we attempt to determine the focal mechanisms of earthquakes with $m_b \geq 4.5$. On the other hand, agencies like JMA and NIED cover a much wider magnitude range together, in comparison to global agencies, offering a more complete coverage in earthquake source mechanisms for Japan. Similar observations can be drawn for agencies BRK, ECX and PNSN covering the seismicity along the western coast of the United States, several European agencies (e.g., MED-RCMT and ZUR-RMT) covering the seismicity of Europe, and agencies RSNC and WEL for Colombia and New Zealand, respectively. This obviously introduces some heterogeneity in the available solutions for different areas and for different magnitude ranges, as a result of the different techniques applied by different agencies for the determination of their source models. Moreover, different agencies report different types of source mechanism and associated parametric data (nodal planes, moment tensor components, principal axes - see also Table 1 for details). More details will be given in Section 3.

All the available source mechanisms are included in the ISC Bulletin. However, users particularly interested in focal mechanisms can search using either a dedicated tool at http://www.isc.ac.uk/iscbulletin/search/fmechanisms/ or webservices at http://www.isc.ac.uk/iscbulletin/search/webservices/fmechanisms/. Search parameters include date, area, magnitude, depth and agency code. Search outputs are available either in a comma separated CSV-like format with one line per mechanism solution, or in QuakeML format. Included in the output are the ISC event identifier, scalar moments, moment tensor components, nodal planes, principal axes and the hypocentre/centroid parameters for each mechanism solution, where applicable. The format is explained in detail at http://www.isc.ac.uk/iscbulletin/search/fmechanisms/csvoutput/.

## 3 Source mechanism variability

The majority of the observed global seismicity is characterized by crustal (shallow) earthquakes which occur as the result of a sudden release of accumulated strain across a seismic fault of finite dimensions. A seismic source whose energy is recorded at stations located at distances of several wavelengths from the source, can be approximated as a point source. The point source model provides a simple and convenient approach in order to simulate the seismic radiation. Nonetheless, for larger earthquakes (for example mega-thrust earthquakes, Tsai et al., 2005; Lentas et al., 2014; Ye et al., 2016), and/or earthquakes observed at distances close to the source (a few kilometers), the point source approximation is not sufficient, and ideally, the rupture propagation history and finite fault characteristics should be taken into account when attempting to model the seismic source. For deep earthquakes, on the other hand, non conventional models have been proposed (e.g., Okal, 2001; Meng et al., 2014), but the point source approximation is also being used for the sake of simplicity and processing consistency with crustal (shallow) earthquakes.

A seismic point source is described in principal by a double couple system of equivalent body forces which are represented by two unit vectors, the normal and slip vectors. These vectors are defined by the orientation of the fault and the direction of slip in terms of the strike, dip and rake angles (e.g., Aki and Richards, 2002). Different techniques follow different concepts for determining the source model of a point source. Some algorithms solve directly for the geometry of a planar fault, meaning the strike, dip and rake of the fault and auxiliary planes assuming a pure double couple mechanism. Other techniques determine the six components of the moment tensor. This is a mathematical representation of the equivalent body forces acting on a seismic point source, and can be decomposed into an isotropic component, a compensated linear vector dipole (CLVD) and a best fitting double couple mechanism describing the geometry of a planar fault.

Source mechanism solutions can be determined by using two main data types: (i) parametric data such as first motion $P$-wave polarities and amplitude ratios, and (ii) waveform data modelling. A vast variety of techniques and algorithms have been developed over the last few decades using different concepts and data. The most robust results are obtained by waveform modelling methods. Even though techniques based on polarities depend strongly on the network geometry and the station azimuthal coverage, they can still be very useful in determining the focal mechanisms of small earthquakes and aftershock sequences using local networks (Shearer, 1998).

Focal mechanisms expressed in terms of strike, dip and rake angles, based on first motion polarities are reported for example by JMA (Nakamura and Mochizuki, 1988), PNSN (FPFIT code by Reasenberg and Oppenheimer, 1985) and ISC (Lentas, 2017). They depend on pre-determined locations and reflect the geometry of the seismic fault at the initial breaking of the rupture. Waveform modelling techniques on the other hand usually consider a group of phases (body waves and/or surface waves) and provide source mechanisms closer to the dominant component of the entire rupture geometry. Their source models can be expressed as strike, dip and rake angles or moment tensor components and they can either be based on a pre-determined location or can be centroid solutions. Centroid based techniques determine the six elements of the moment tensor, the centroid location (latitude, longitude and depth of a point source) and the origin time simultaneously. Waveform modelling methods such as the SCARDEC technique (Vallée et al., 2010), whose source models are reported to the ISC by agency IPGP, make

use of body-wave phases and the NEIC location. The obtained source models are expressed as strike, dip and rake angles of a pure double couple source. Source models of other waveform modelling techniques based on a pre-determined hypocentre location are also routinely reported in the ISC Bulletin, such as the NIED solutions for the Japanese area (Fukuyama and Kawai, 1998). In this case the reported source models are expressed in terms of the six elements of the moment tensor and the

best fitting double couple solution. Centroid based techniques like the GCMT (Dziewonski et al., 1981; Ekström et al., 2012), MED-RCMT (Pondrelli et al., 2011), ZUR-RMT (Braunmiller et al., 2002) and others applied by the NEIC (Hayes et al., 2009; Benz and Herrmann, 2014) provide centroid locations, moment tensor components and best fitting double couple solutions.

Source models for all the above mentioned techniques can be found in the ISC Bulletin (see Fig. 2). Moreover, taking into consideration additional differences in velocity models, station distribution and observations in different waveform frequency

bands that are being used by different techniques, some variation among source mechanisms reported by different agencies for the same seismic event is to be expected.

Figure 5 shows the frequency distribution of available source mechanism solutions per event in the ISC Bulletin, and the frequency distribution of maximum intra-event rotation angle for the events having at least two reported mechanism solutions. The rotation angle describes the transformation of a double couple mechanism into another arbitrary mechanism through

3-D rotations (Kagan, 1991). To further visualise this it is worth noting that the rotation angle between two double couple mechanism solutions can vary between 0° and 120°, where 0° corresponds to perfect match and 120° describes absolute mismatch which physically translates into mechanism solutions showing for example perpendicular strike orientations and/or conflicting fault types (e.g., normal compared to thrust, or normal compared to pure strike slip, and so on). Consequently, in order to determine the maximum intra-event rotation angle we list all the available best fitting double couple source mechanism

solutions for each event and we calculate rotation angles for all possible pairs. We then pick the maximum value as describing the greatest mismatch between the available mechanisms, and hence, the extent of expected differences in strike, dip and rake (see for example Fig. 1 in Cesca et al., 2013).

A substantial number of earthquakes in the ISC Bulletin ($\sim$40%) show intra-event rotation angles between 10° and 25° (see Figure 5). Cases showing large differences, above 40°, occasionally occur and can be partly explained by earthquakes showing

complex rupture, such as the 2002 November 3, $M_W$ 7.9, Denali Central Alaska earthquake (e.g., Ozacar et al., 2003) and the doublet 2012 December 7, $M_W$ 7.2, east coast of Honshu earthquake (e.g., Lay et al., 2013) which show intra-event rotation angles up to 60° and 90° respectively, or comparing automatic source mechanism solutions such as the 2015 May 25, $M_W$ 5.2, eastern Honshu earthquake which shows intra-event rotation angles up to $\sim$90°.

In the case of the Denali event (http://www.isc.ac.uk/cgi-bin/web-db-v4?event_id=6123395&out_format=IMS1.0&request=

COMPREHENSIVE), the earthquake started as a thrust event and then ruptured along the curved strike-slip Denali fault (Ozacar et al., 2003). Similarly, Lay et al. (2013) suggested the case of a doublet event that began with a $M_W$ 7.2, thrust earthquake (http://www.isc.ac.uk/cgi-bin/web-db-v4?event_id=607215270&out_format=IMS1.0&request=COMPREHENSIVE) and followed by a $M_W$ 7.1-7.2, normal-faulting earthquake (http://www.isc.ac.uk/cgi-bin/web-db-v4?event_id=602005586&out_format=IMS1.0&request=COMPREHENSIVE) for the case of the 2012 east coast of Honshu earthquake. Since the routinely

reported mechanism solutions for these complex events are based on the point source approximation, it is no surprise that

different episodes of rupture are captured by different methods using different data. For example, the Global CMT, by mainly using long period surface waves (and in some cases long period body waves), has detected the strike-slip nature of the Denali earthquake. In contrast, NEIC source mechanism captured the initial stage of the rupture and shows evidence of thrust faulting. Note that large intra-event rotation angles can occur also for moderate earthquake, such as for the 2015 May 25, $M_W$ 5.2, eastern Honshu earthquake. Indeed, for this event the intra-event rotation angles are up to ~90°. It is likely that such variability is due to differences in the methods and data applied (first-motion by JMA and ISC versus waveform modelling by NIED and GCMT) rather than to rupture complexity. Substantial intra-event differences are also very common as a result of multiple solutions reported by PNSN for small earthquakes with poorly constrained source mechanisms, such as the 1981 February 11, $M_d$ 2.5, Washington earthquake where the maximum intra-event rotation angle can be as high as 100°. This is very common in the case of PNSN reported mechanisms due to the first motion technique that is being used which provides multiple solutions if the data is not adequate for the determination of a single well constrained solution. Quality characterization found in the comments in the online ISC Bulletin can help users to identify the most robust mechanism solution among multiple PNSN provided mechanisms for the same event.

Intra-event rotation angles are not currently calculated and published in a systematic way in the ISC Bulletin, and hence, the identification of cases of substantial differences in reported source models is not part of the ISC standard procedures. Researchers who are interested in earthquake source model validations and assessment are encouraged to make use of the ISC Bulletin for this purpose and apply their own schemes. As already mentioned, the source models in the ISC Bulletin are not reviewed by the analysts. However, we frequently carry out health checks of the bulletin, namely, removing duplicates and checking for consistency between moment tensors and best fitting double couple solutions, or moment tensors and principal axes, or double couple solutions and principal axes and so on. This has resulted in correcting a few cases (less than 1% of the entire bulletin) where either typos from the data providers or bugs in the database parsing algorithms have been identified and fixed. As part of this process we did not detect systematic inconsistencies between source models provided by different agencies for the same seismic event that has to do with the type of source mechanism, or depth, or a specific region. The most common case in this respect is source mechanisms provided by PNSN for small magnitude events along the western coast of United States due to the type of methodology that is being applied which determines multiple solutions when the data quality is poor.

Source mechanisms are also used in a variety of studies (e.g., tectonics, stress patterns, cluster analysis) aiming at characterizing an event or a set of events by the fault style. Several classifications of fault styles are available in the literature (see, e.g., Célérier, 2010, for an overview). For this work we adopt the classification proposed by Zoback (1992) for the World Stress Map Project (http://www.world-stress-map.org/). With such classification an event is assigned one of the following fault styles: thrust, normal, strike-slip, normal with strike-slip component, thrust with strike-slip component and undefined for events not fitting in any of the previous categories (similarly to the "odd" group in Frohlich, 1992). Figure 6 shows the annual number of earthquakes grouped according to Zoback (1992) and the sole intent of the figure is to showcase one of the possible uses for the source mechanisms in the ISC Bulletin.

Considering the source mechanisms in the ISC Bulletin, as discussed so far, it is obvious that for events with only one mechanism available the fault style is easily assigned. The same also applies to events with multiple solutions all of the same fault style (e.g., http://www.isc.ac.uk/cgi-bin/web-db-v4?event_id=3021752&out_format=IMS1.0&request=COMPREHENSIVE). However, for events with more than one solution it is not always possible to assign a fault style with the solutions at hand. This happens, for example, when an event has two source mechanisms and one being thrust and the other normal (e.g., http://www.isc.ac.uk/cgi-bin/web-db-v4?event_id=602214316&out_format=IMS1.0&request=COMPREHENSIVE). Similarly, if an event has multiple solutions, we may have source mechanisms falling into more than two fault styles (e.g., http://www.isc.ac.uk/cgi-bin/web-db-v4?event_id=602431903&out_format=IMS1.0&request=COMPREHENSIVE) or without a unique maximum in the number of source mechanisms belonging to a fault style (e.g., http://www.isc.ac.uk/cgi-bin/web-db-v4?event_id=602945524&out_format=IMS1.0&request=COMPREHENSIVE). In such cases we do not assign a fault style to an event. If, instead, out of the fault style distribution within an event there is a more recurrent fault style (e.g., http://www.isc.ac.uk/cgi-bin/web-db-v4?event_id=2944860&out_format=IMS1.0&request=COMPREHENSIVE), we still assign a fault style to the event. Therefore, in Figure 6 we also show the "Discrepant" category for those events where we could not assign a specific fault style. Note that also complex earthquakes, such as the ones previously mentioned, may fall into this category. The annual percentage of events falling into the "Discrepant" category is usually between 0 and 5%, with a maximum of 8% in 2000. The occurrence of such "Discrepant" events should not discourage the use of source mechanisms from the ISC Bulletin. Indeed, this category of events can highlight complex events or events for which more studies are needed.

Not surprisingly, Figure 6 shows how the thrust earthquakes dominate the annual occurrences, with the only exceptions for year 2000 (significant strike-slip aftershock sequence following the 2000 Tottori, Japan, earthquake) and 2011 (aftershocks of the 2011 Tohoku earthquake, due to the stress field change, were characterized by many normal-fault earthquakes, as shown, e.g., by Hasegawa et al., 2012). Such use of the source mechanisms in the ISC Bulletin can be used for event characterizations and applied, e.g, to studies concerning regional stress patterns (e.g., Balfour et al., 2011), magnitude (e.g., Lomax and Michelini, 2009), tsunami/tsunamigenic earthquakes (e.g., Okal and Newman, 2001), and other seismological fields.

## 4 Data availability

The earthquake source mechanisms summarized in this work are freely available within the ISC Bulletin websearch at http://doi.org/10.31905/D808B830. All data used in this paper are maintained at the ISC (www.isc.ac.uk, last accessed 15 March 2019).

## 5 Summary and conclusions

The ISC offers the most comprehensive bulletin of global seismicity in terms of hypocentre solutions, phase arrivals, magnitudes and amplitude measurements. In this paper we present an additional aspect of the ISC Bulletin, namely its source

30    mechanism content and the opportunity for ISC users to complement the source mechanism information with all other data included in the ISC Bulletin.

The ISC has a mandate to collect as much parametric data as possible from various sources around the world and make it freely available to the seismological as well as to a broad geoscience community. As a result, the magnitude range of earthquakes covered by available mechanism solutions is larger than individual global catalogues such as GCMT or NEIC. However, this feature inevitably leads to a higher heterogeneity in the solutions due to different methods adopted by each

provider. Thus, users are advised to be aware of the techniques being used in the computations of the various source models in the ISC Bulletin. For example, centroid based mechanism solutions should be used together with centroid locations, since: (i) both the centroid mechanism and centroid location are parts of the same output, and (ii) substantial differences may exist among centroid locations and standard hypocentre locations fitting the observed phase arrival times of body waves. To facilitate this the CSV format provided by the online ISC Bulletin indicates whether a mechanism solution is centroid or hypocentre. Source

mechanisms obtained from pre-determined standard hypocentre locations can be used together with the provider's hypocentre solution or the prime hypocentre solution in the ISC Bulletin. However, large differences in depth may be present in some cases among the prime hypocentre solution and the solution provided by the mechanism's agency. Moreover, information regarding the quality of the obtained mechanism solutions such as the number of stations being used and/or errors in the obtained source models is also provided in the ISC Bulletin where available from the reporting agency. The latter is more common in moment

tensor solutions. Detailed quality information is routinely provided for the ISC focal mechanism solutions both in the comments section in the online ISC Bulletin as well as by clicking on the "ISC Focal Mechanism" logo on the top of the online bulletin.

In the case of multiple available source mechanism solutions for the same event the question "which source model should I pick for my research from the ISC Bulletin" arises from a researcher's practical point of view. Since the source model determination problem is not fully resolved and the point source approximation is still the standard in routine earthquake

mechanism representation, unfortunately there is no easy and straightforward answer to this question. The purpose of this paper is to give insights to the content of the ISC Bulletin regarding the availability of earthquake source models for seismologists, and further highlight the complexity of the earthquake source and the associated point source models for the broad geoscience community. Thus, only general suggestions can be given without attempting or willing to discriminate the data in the ISC Bulletin in right or wrong. Users of the bulletin should keep in mind that different techniques are based on different data

(first motion polarities, body wave amplitude ratios, body/surface wave modelling) and as a result they weight differently various episodes in the rupture history which ultimately can have a strong effect on the final solution. For example, slip takes place across seismic faults that are not necessarily planar, but their orientation can vary with length. Moreover, the use of local/regional or teleseismic data can be another component of source model variations, and in conjunction with the uncertainties in velocity models that are being used to simulate the wave propagation in the Earth's interior it is advisable to

take into consideration the details of the methodologies being used by different source mechanism providers.

In all cases, the ISC will not deprecate solutions without being revised by the data provider. The ISC cannot encourage the use of source models of one data provider over another. We try to include as much information as possible in the comments section of the online ISC Bulletin regarding each mechanism solution. Researchers are free to make use of any source models

they might think that fit their research and to facilitate this we advise users to pay attention to quality information provided in the comments section, namely, the number of stations and components being used, the azimuthal gap and so on. Another tool for selecting the most appropriate source model is the association of the mechanism solution with the type of location being used (hypocentre/centroid) as stated above. In this paper we briefly summarized the methods being used by major model reporters. However, users still need to identify the most appropriate source models for their research.

Similar to the variation of hypocentre solutions in the ISC Bulletin, multiple source mechanisms for the same seismic event, when available, can provide a measure of the posterior uncertainties with respect to data errors and modelling techniques. Despite variations in methods and data used to compute the solutions, we showed that in most cases there is a good agreement among multiple solutions provided by different agencies. The intra-event variability in the ISC Bulletin was quantified by the maximum rotation angle and is well constrained up to $20°$, which corresponds to a similarity coefficient of $\sim 80\%$ (Cesca et al.,

2013). As already mentioned above both moment tensors and focal mechanism solutions are included in the ISC Bulletin. Since we needed to compare results obtained from these fundamentally different concepts, obtained by different techniques, we decided to focus on pure double couple and moment tensor best fitting double couple mechanisms, which describe the geometry of the seismic source. For this reason the use of the rotation angle (Kagan, 1991) was selected. Different classification techniques and metrics could be applied (i.e., Helffrich, 1997; Frohlich and Davis, 1999) but for the purpose of the current

article the rotation angle is considered to be an adequate metric of source mechanism variability. Similarly, by applying Zoback (1992) fault styles classification, we observe a large intra-event variability for up to 8% of the earthquakes per year.

    Source mechanisms are currently not reviewed by ISC analysts, and the user should pick, if required, the preferred one when multiple solutions are available for an event. The ISC values all agencies reporting source mechanism solutions and encourages new ones to submit theirs. In parallel, we recommend researchers make a more systematic use of the earthquake

source mechanisms in the ISC Bulletin in future studies.

*Author contributions.*   KL was the leading author of the paper and compiled most of the figures. DDG vetted the data for the magnitude and fault styles plots, JH maintains the database and webservices and DAS obtained the funding for the work and established and maintained connections with many data providers. All authors contributed to the manuscript and approved the final version.

*Competing interests.*   The authors declare that they have no conflict of interest.

*Acknowledgements.*   The authors wish to thank two anonymous Reviewers for their comments and suggestions which hepled to improve this manuscript. The work done at the ISC is possible thanks to the support of its members (www.isc.ac.uk/members/,) and sponsors (www.isc.ac.uk/sponsors/). Work partially funded by NSF Grants 1417970 and 1811737, and a USGS Award G18AP00035. Figures were drawn using the Generic Mapping Tools (Wessel et al., 2013) and the Matplotlib python library (Hunter, 2007).

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

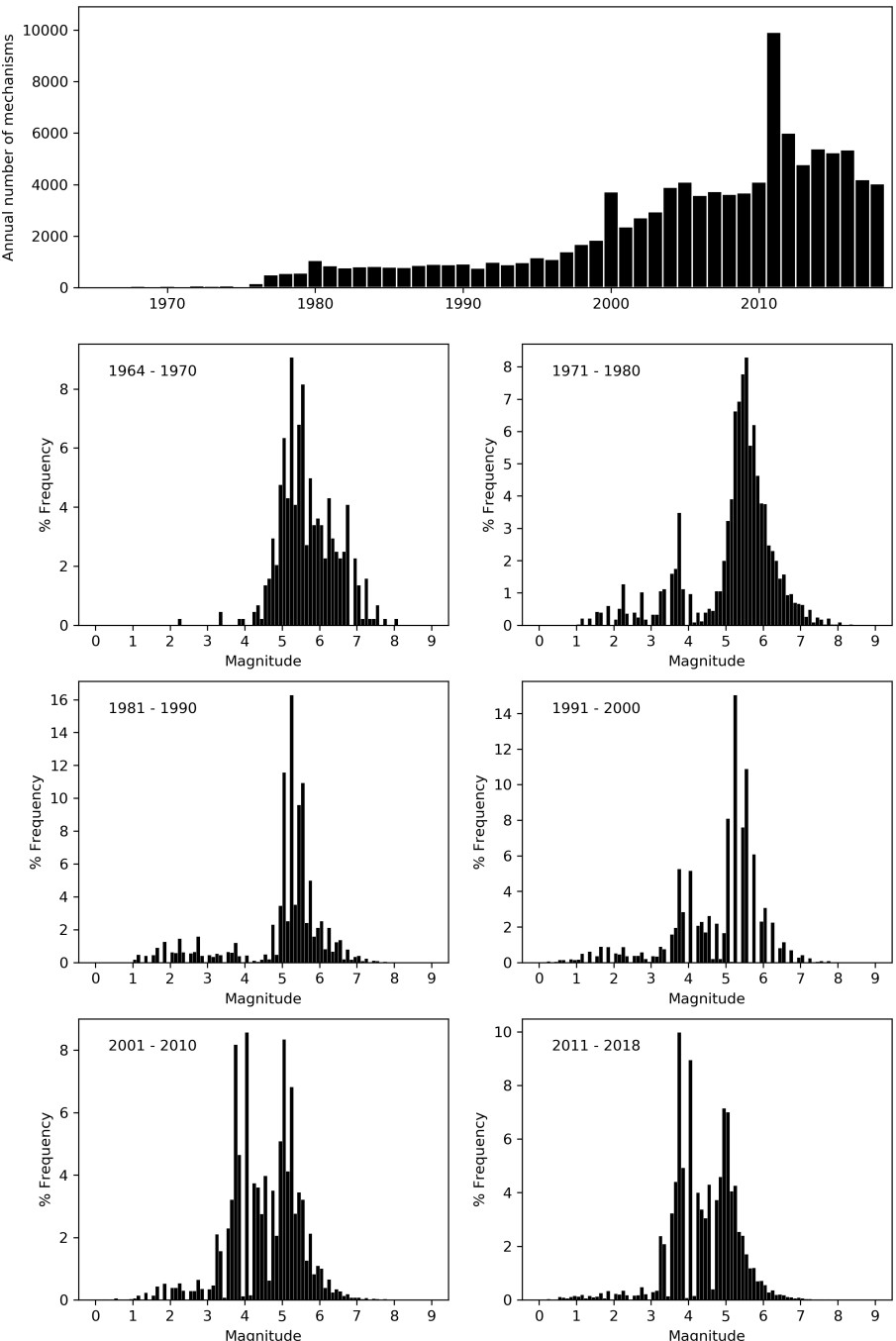

**Figure 1.** Number of source mechanisms in the ISC Bulletin from January 1964 to December 2018 and normalized frequency - magnitude distributions with respect to different time periods. A magnitude value for each event with an associated source mechanism is selected following the scheme described in Di Giacomo and Storchak (2015). Note the high peak in the top subplot in 2011 which is associated with the 2011 Tohoku earthquake aftershock sequence. A slight dip for year 2018 is only apparent as the data collection is not yet complete.

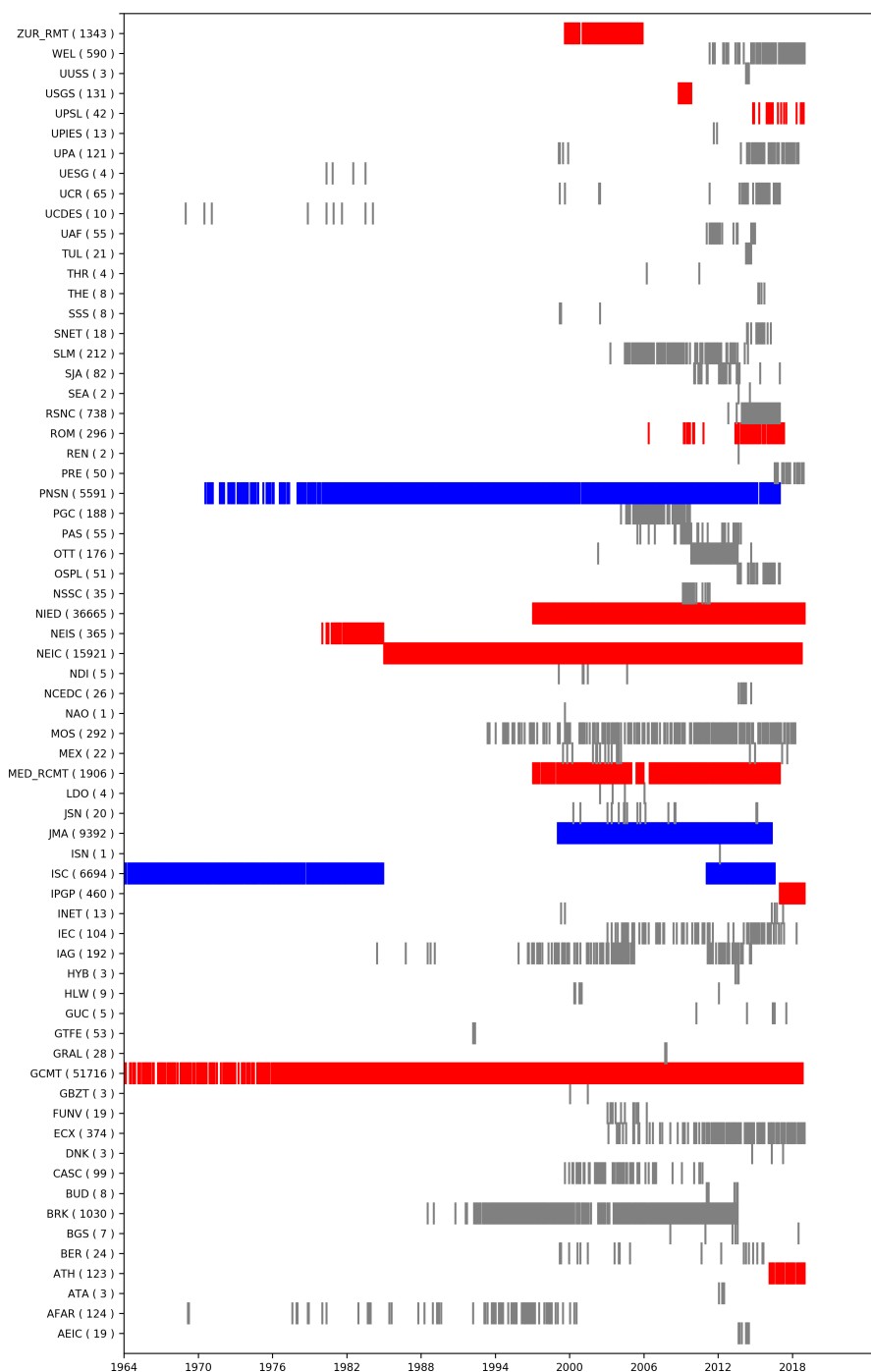

**Figure 2.** Time distribution of source mechanisms in the ISC Bulletin (January 1964 – December 2018) reported by different agencies. The numbers in brackets next to each agency code shows the total number of reported source mechanism solutions. Red lines indicate waveform inversion techniques, blue lines indicate first motion polarity techniques and grey colour shows cases where there is no information available on the techniques being used or we could not verify them. A detailed list of the reporting agencies is shown in Table 1.

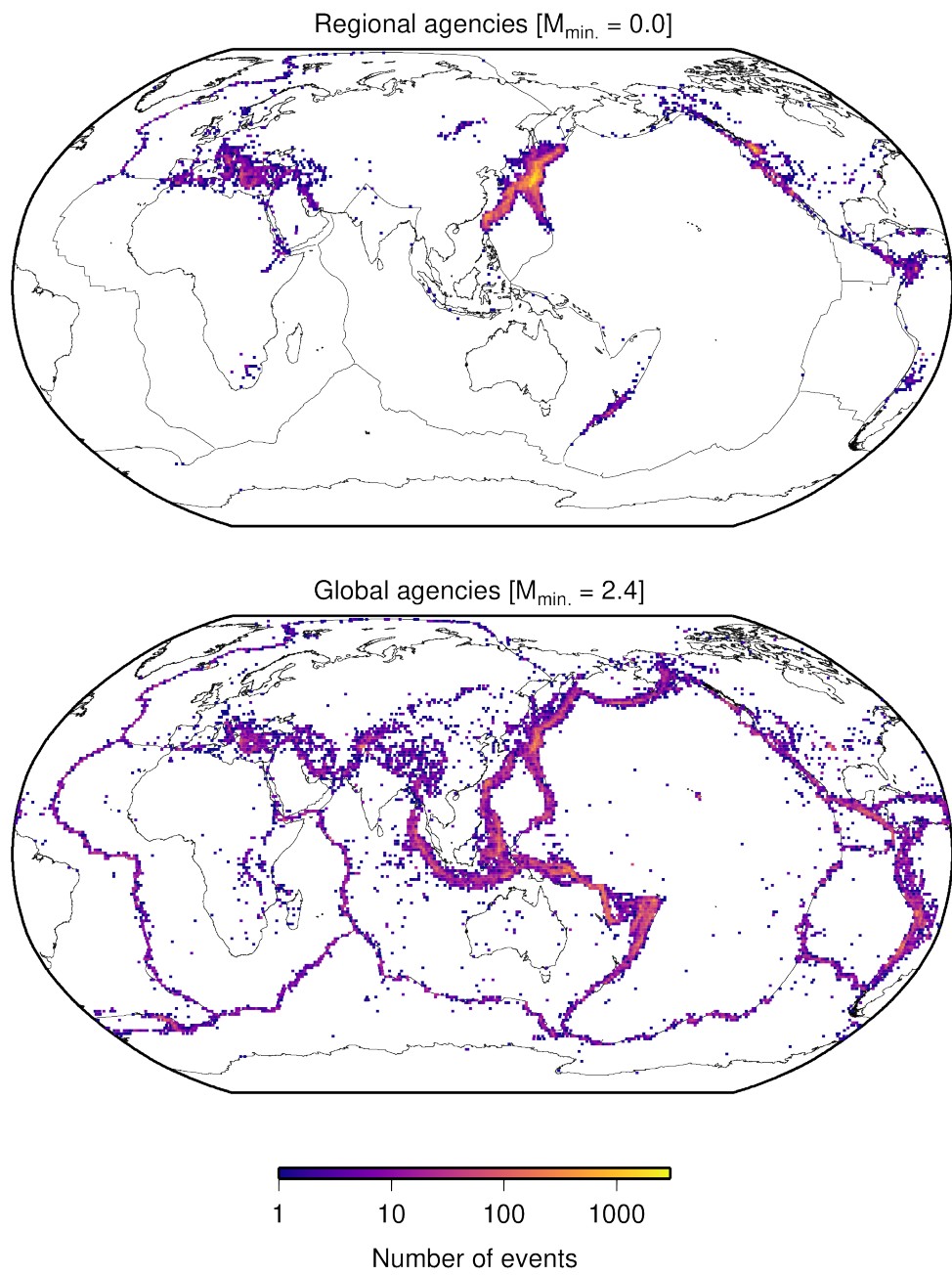

**Figure 3.** Global maps showing the number of events with at least one source mechanism reported in the ISC Bulletin for the time period from January 1964 to December 2018, in a 1° by 1° grid, for mechanism solutions reported by local and regional agencies (top), and global agencies (bottom, [GCMT, HRVD, IPGP, ISC, MOS, NEIC, NEIS]). Minimum magnitude ($M_{min.}$) following the scheme described in Di Giacomo and Storchak (2015) is shown on top of each map.

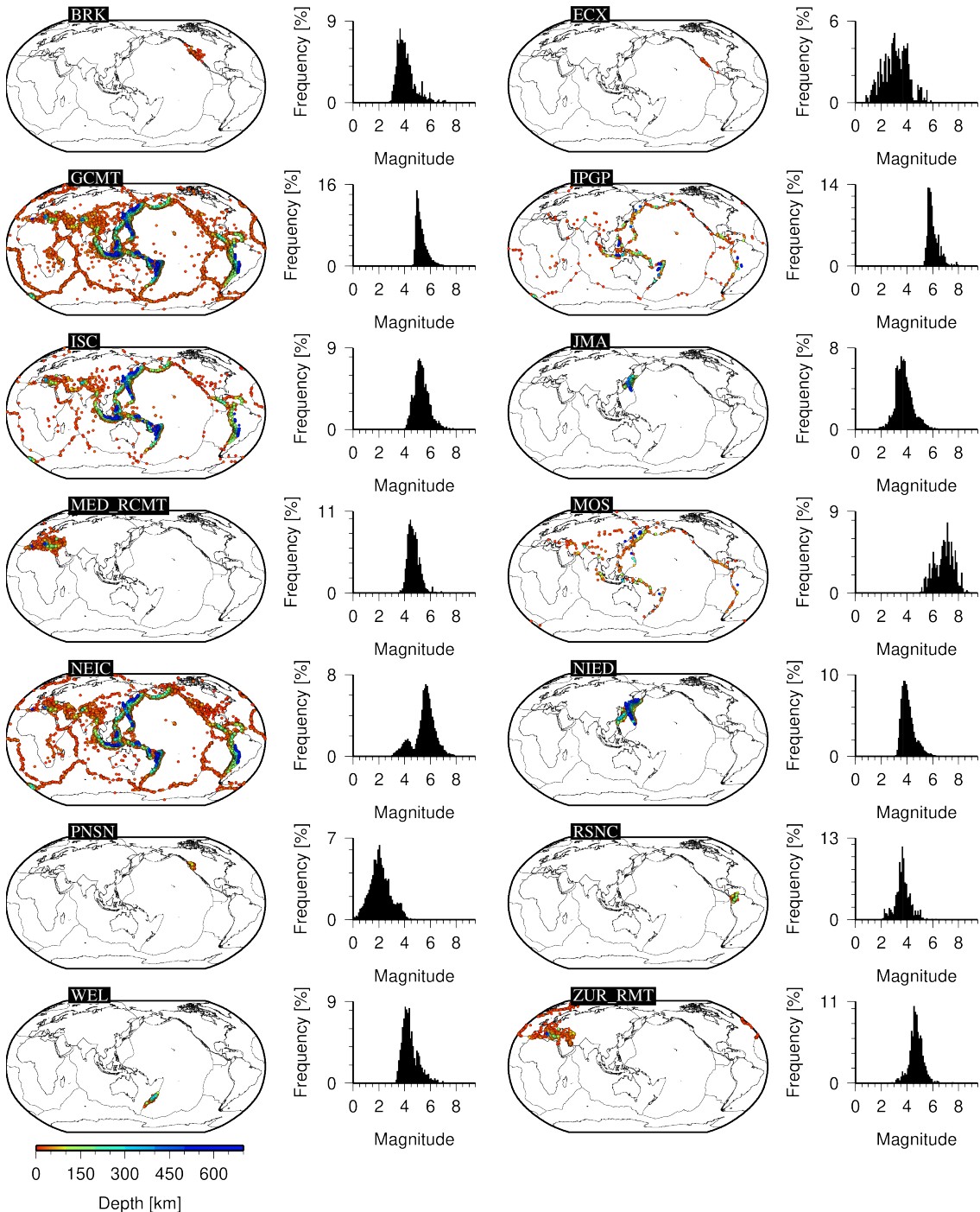

**Figure 4.** Geographical and frequency - magnitude distribution (following the scheme described in Di Giacomo and Storchak (2015)) of earthquakes with source mechanisms reported by the agencies which systematically send their mechanism solutions to the ISC. Locations on the maps are colour coded by depth. The agency codes are shown on top of each map and their details can be found in Table 1.

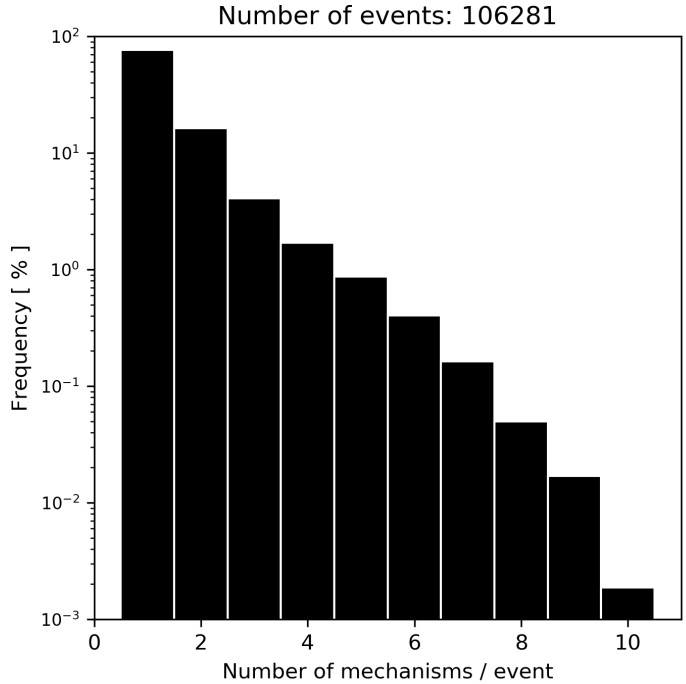

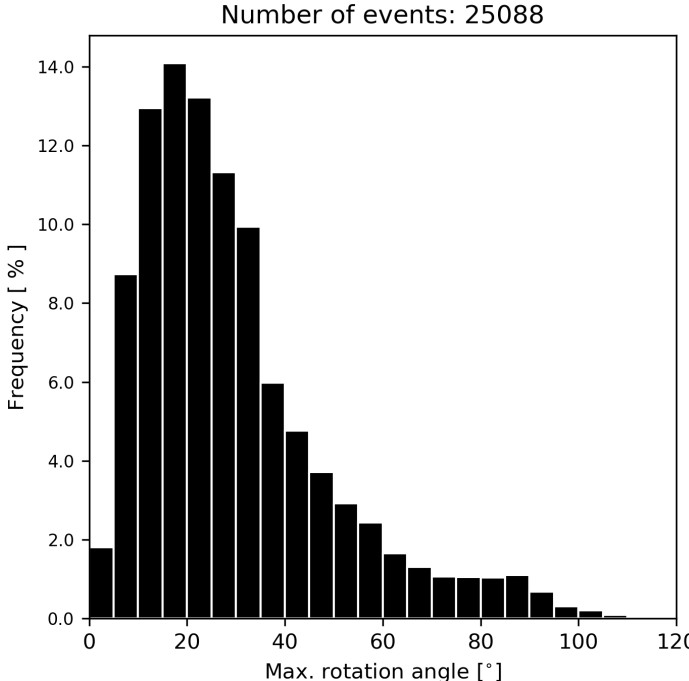

**Figure 5.** Frequency distribution of available source mechanisms per event in the ISC Bulletin for the time period from January 1964 to December 2018 (top), and intra-event maximum rotation angle frequency distribution for the seismic events having at least two mechanism solutions available (bottom).

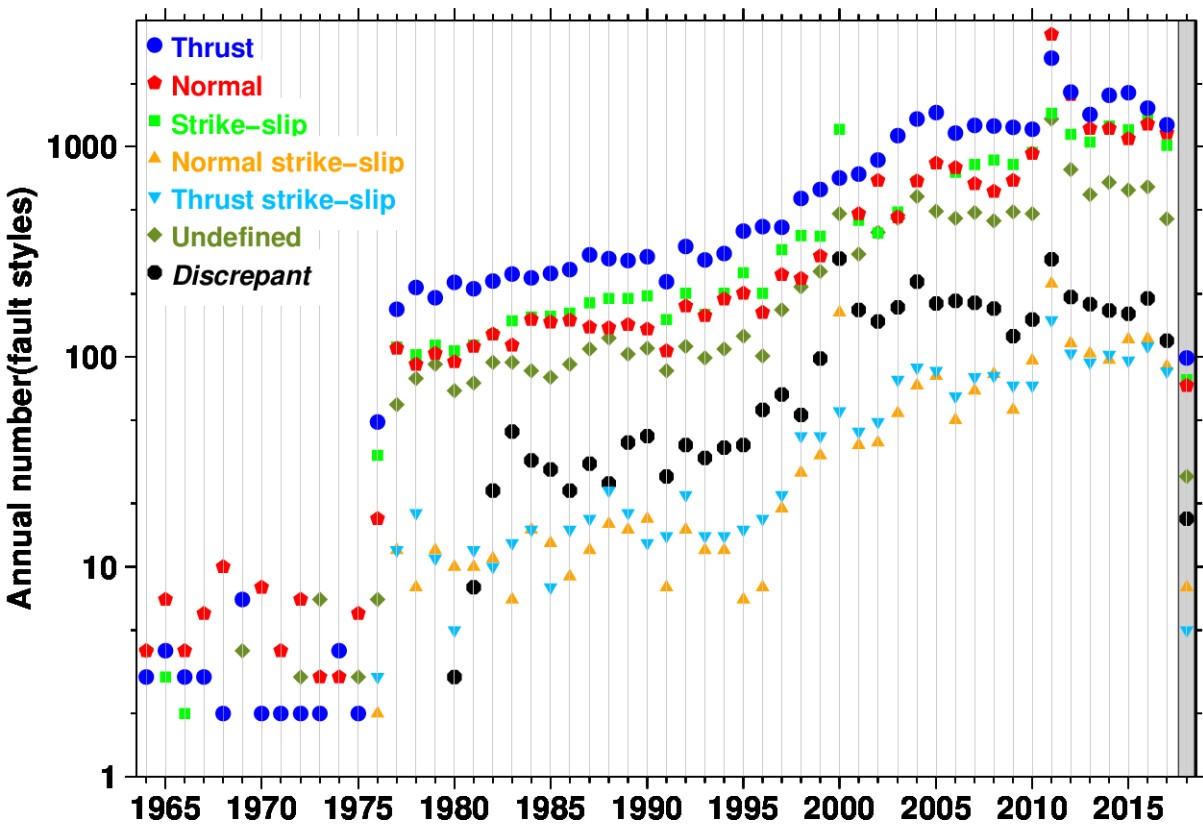

**Figure 6.** Annual number of source mechanisms grouped by Zoback (1992) fault styles. Dark blue circles, red pentagons, green squares, orange triangles, light blue inverted triangles and olive diamonds represent, respectively, thrust, normal, strike-slip, normal with strike-slip component, thrust with strike-slip component and undefined earthquakes. Black octagons are earthquakes for which the intra-variability of the source mechanisms does not allow us to assign an earthquake to a fault style. These are earthquakes with large rotation angles (see Figure 5). The percentage of such earthquakes goes up to 8% of earthquakes per year. The grey area for year 2018 denotes that the data collection is not yet complete. See main text for details.

**Table 1.** Agencies reporting source mechanism solutions at the ISC for the time period from January 1964 to December 2018. The cross symbol (x) denotes the type of parametric data that is reported from each agency to the ISC.

| Agency code | Name | Country | Mag. range | Nodal planes | Moment tensor | Principal axes |
|---|---|---|---|---|---|---|
| AEIC | Alaska Earthquake Information Center | U.S.A. | 3.6–5.0 | | X | |
| AFAR | The Afar Depression: Interpretation of the 1960-2000 Earthquakes | Israel | 3.8–6.6 | X | | |
| ATA | The Earthquake Research Center Ataturk University | Turkey | 3.9–4.5 | X | | |
| ATH | National Observatory of Athens | Greece | 3.7–5.5 | X | | |
| BER | University of Bergen | Norway | 1.1–4.0 | X | | |
| BGS | British Geological Survey | United Kingdom | 1.2–4.6 | X | | |
| BRK | Berkeley Seismological Laboratory | U.S.A. | 2.9–7.2 | X | | |
| BUD | Geodetic and Geophysical Research Institute | Hungary | 2.0–4.7 | X | X | X |
| CASC | Central American Seismic Center | Costa Rica | 3.1–6.5 | X | | |
| DNK | Geological Survey of Denmark and Greenland | Denmark | 2.5–4.6 | X | | |
| ECX | Centro de Investigación Científica y de Educación Superior de Ensenada | Mexico | 0.9–5.8 | X | | |
| FUNV | Fundación Venezolana de Investigaciones Sismológicas | Venezuela | 1.9–4.2 | X | | |
| GBZT | Marmara Research Center | Turkey | 3.7–5.4 | X | | |
| GCMT | The Global CMT Project | U.S.A. | 4.0–9.1 | X | X | X |
| GRAL | National Council for Scientific Research | Lebanon | 1.4–3.3 | X | | |
| GTFE | German Task Force for Earthquakes | Germany | 2.2–4.6 | X | | |
| GUC | Centro Sismológico Nacional, Universidad de Chile | Chile | 2.9–4.3 | X | | |
| HLW | National Research Institute of Astronomy and Geophysics | Egypt | 2.5–4.9 | X | | X |
| HYB | National Geophysical Research Institute | India | 3.0–3.8 | X | | |
| IAG | Instituto Andaluz de Geofisica | Spain | 3.2–6.8 | X | X | X |
| IEC | Institute of the Earth Crust, SB RAS | Russia | 3.3–6.3 | X | | X |
| INET | Instituto Nicaraguense de Estudios Territoriales - INETER | Nicaragua | 3.5–6.0 | X | | |
| IPGP | Institut de Physique du Globe de Paris | France | 5.2–8.2 | X | | |
| ISC | International Seismological Centre | United Kingdom | 3.6–9.1 | X | | X |
| ISN | Iraqi Meteorological and Seismology Organisation | Iraq | 5.0 | X | | |
| JMA | Japan Meteorological Agency | Japan | 1.6–8.3 | X | | X |
| JSN | Jamaica Seismic Network | Jamaica | 2.5–5.2 | X | | |
| LDO | Lamont-Doherty Earth Observatory | U.S.A. | 3.5–4.6 | X | X | |
| MED RCMT | MedNet Regional Centroid - Moment Tensors | Italy | 3.6–6.9 | X | X | X |
| MEX | Instituto de Geofísica de la UNAM | Mexico | 2.9–7.0 | X | | |
| MOS | Geophysical Survey of Russian Academy of Sciences | Russia | 5.1–9.1 | X | | X |
| NAO | Stiftelsen NORSAR | Norway | 2.5 | X | | |
| NCEDC | Northern California Earthquake Data Center | U.S.A. | 3.0–5.2 | | X | |
| NDI | National Centre for Seismology of the Ministry of Earth Sciences of India | India | 3.5–7.7 | X | | |
| NEIC | National Earthquake Information Center | U.S.A. | 2.4–9.1 | X | X | X |
| NEIS | National Earthquake Information Service | U.S.A. | 5.3–7.7 | X | X | X |
| NIED | National Research Institute for Earth Science and Disaster Prevention | Japan | 3.1–9.1 | X | X | |
| NSSC | National Syrian Seismological Center | Syria | 1.6–4.6 | X | | |
| OSPL | Observatorio Sismologico Politecnico Loyola | Dominican Republic | 2.2–5.8 | X | | |
| OTT | Canadian Hazards Information Service, Natural Resources Canada | Canada | 3.6–7.8 | X | | |
| PAS | California Institute of Technology | U.S.A. | 3.3–7.2 | X | X | X |
| PGC | Pacific Geoscience Centre | Canada | 3.5–6.6 | X | X | X |
| PNSN | Pacific Northwest Seismic Network | U.S.A. | 0.0–6.8 | X | | X |
| PRE | Council for Geoscience | South Africa | 2.2–6.5 | X | | |
| REN | MacKay School of Mines | U.S.A. | 3.3–4.2 | | X | |
| ROM | Istituto Nazionale di Geofisica e Vulcanologia | Italy | 3.1–6.6 | X | X | |
| RSNC | Red Sismológica Nacional de Colombia | Colombia | 1.6–6.3 | X | | |
| SEA | Geophysics Program AK-50 | U.S.A. | 3.2–3.3 | | X | |
| SJA | Instituto Nacional de Prevención Sísmica | Argentina | 2.6–5.9 | X | | |
| SLM | Saint Louis University | U.S.A. | 2.2–6.0 | X | X | X |
| SNET | Servicio Nacional de Estudios Territoriales | El Salvador | 1.3–5.6 | X | | |
| SSS | Centro de Estudios y Investigaciones Geotecnicas del San Salvador | El Salvador | 3.6–6.0 | X | | |
| THE | Department of Geophysics, Aristotle University of Thessaloniki | Greece | 4.2–6.1 | X | | |
| THR | International Institute of Earthquake Engineering and Seismology (IIEES) | Iran | 4.5–6.1 | X | | |
| TUL | Oklahoma Geological Survey | U.S.A. | 2.9–3.7 | | X | |
| UAF | Department of Geosciences | U.S.A. | 0.5–5.2 | X | X | |
| UCDES | Department of Earth Sciences | United Kingdom | 5.5–6.7 | X | | |
| UCR | Sección de Sismología, Vulcanología y Exploración Geofísica | Costa Rica | 2.6–6.0 | X | | |
| UESG | School of Geosciences | United Kingdom | 5.7–6.2 | X | | |
| UPA | Universidad de Panama | Panama | 1.8–6.6 | X | | |
| UPIES | Institute of Earth and Environmental Science | Germany | 4.3–6.4 | X | X | |
| UPSL | University of Patras, Department of Geology | Greece | 3.3–6.8 | X | | |
| USGS | United States Geological Survey | U.S.A. | 3.2–7.8 | X | X | X |
| UUSS | The University of Utah Seismograph Stations | U.S.A. | 3.3–3.8 | | X | |
| WEL | Institute of Geological and Nuclear Sciences | New Zealand | 3.4–7.1 | X | | X |
| ZUR RMT | Zurich Moment Tensors | Switzerland | 2.8–7.6 | X | X | X |