# Peer review of "The ISC Bulletin as a comprehensive source of earthquake source mechanisms"

_Earth System Science Data, 2018_

## Referee Comment (RC1) · Anonymous Referee #1 · 17 Dec 2018

Review of the manuscript "The ISC Bulletin as a comprehensive source of earthquake source mechanisms" by K. Lentas and coauthors, submitted to Earth System Science Data Discussion.

This compact paper describes the recompilation of global earthquakes source parameters (focal mechanisms and moment tensor solutions) included in the ISC Bulletin.

I believe that a description of this useful source of information for seismologists deserved to be published, but at the same time I found the paper unclear and incomplete and I have a number of important comments which should be considered in a revised manuscript.

I suggest to reconsider the publication after major revision.

[Figure]

I list below some main comments and several minor ones.

Main comments

1. On own mechanisms and reported ones Discussing the seismic source catalog at ISC, one wonders why only reported focal mechanisms are discussed in this paper, and not also those computed by ISC itself. Please clarify. This appears first in the abstract, where you mention that "the main sources of focal mechanisms in the ISC Bulletin are reported solutions ... and ISC computed focal mechanisms..." but then you "focus only on the reported mechanisms". To me this makes no sense, since the aim of the paper is to discuss ISC as a source of information, but you present only part of it.

2. ISC and other global sources When discussing ISC as a source of information for focal mechanisms and moment tensors on a global base, more reference should be given to those sites/Institutes which provided such information for a long time, and highlight differences (e.g. in terms of geographical and magnitude targets, number of reported solutions, time frame, and provided information). This implies Global CMT at first, but also other sources. Such comparative discussion should point out main differences and complementarity. An interesting aspect, here, is that ISC recompiles and mirrors a number of different solutions, whereas e.g. Global CMT is basically performing its own inversion and provide one solution per event. This has potential positive and negative aspects, as I discuss below.

3. Section 3 restructuring The paragraphs at Pag.3 - lines 20-32 is unclear. I suggest to reformulate these paragraphs. I think the first part of Section 3 reads fine and focus on differences in terms of fitting approach (polarity/amplitudes/amplitude ratios vs waveform based approaches). The continuation is less clear. I would suggest to restructure the text as follow: first discuss point source approximation, then come to point source models (focal mechanisms, moment tensors), then to the approaches to resolve them (polarity, amplitude ratios, waveforms etc.). Here you should also discuss

differences among moment tensors and centroid moment tensors – which are recalled later in the paper. And link focal mechanisms to hypocenters and evtl. moment tensors to centroids, as you mention later differences among hypocenters and centroids.

4. From multiple solutions to interpretations The second part of Section 3 is dedicated to describe a peculiar feature of the catalog, which is the presence – for some events – of multiple solutions. This is potentially a nice feature, but also makes the life of a user more difficult, as it becomes unclear to what solution to believe a) I think authors could discuss more the considered case studies (P. 4 – L. 8-12) b) Suggest if possible, what approach to follow to identify cases of inconsistent solutions (and mention if rotation angles are provided in the catalog or not) and ideally how to choose a reference mechanism c) The discussion of mechanisms variability is very poor, and miss to investigate what are typical reasons for mechanism inconsistency for some events. Are these events located in some specific regions, e.g. where to little or to noisy data are available? Is there a relation to magnitude? Or to depth? Do they involve all type of focal mechanisms? d) Very obscure is how authors go from the multiple solutions to Figure 5. I think there is a step missing in this part of the text. How do you obtain a fault style? How do you handle cases where multiple solutions fall into different styles? How (and if) do you choose one solution out of the multiple solutions.

5. Figures Figures can be improved as follow: a) Figure 1 could be improved by also showing the temporal evolution of target magnitudes, or an histograms discussing magnitude distribution b) Figure 2: institute acronyms should be listed somewhere, perhaps in an additional table. c) Figure 3 would be more readable if a title is included in the figure for the upper and bottom panels (i.e. Regional agencies M>xx, Global agencies M> xx) d) Figure 5: Add a symbol legend, so that the figure is better readable. e) Figure 5 caption: what are "undefined earthquakes"?

Other minor comments

1. Clarify the meaning of 90% at Pag.1 - Line 8, or reformulate the text.

2. P. 4 – Line 5: maximum intra-event rotation: I understand what you mean, but feel this needs a better explanation for a broader audience. Please, first introduce this concept clearly.

3. P. 4 – Line 14: "... can be as high as 100°". I think this is not very helpful for a reader, as one cannot get a feeling for this number. Perhaps you could mention what is the maximum possible value for the rotation angle.

4. P. 4 – Line 21: what are "earthquakes for which we could not assign a fault style"? How is a fault style assigned? Not clear at all.

5. P. 5, Line 8 "be aware of the techniques being used". I think this becomes important especially for the cases where multiple, inconsistent solutions are reported. A user would probably rely on a certain technique or based on the data amount used for the inversion, or the azimuthal coverage. IS the Bulletin reporting all these details? If not, the user has no options to be aware...

6. P. 5, Line 12 "centroid or not" should be "centroid or hypocenter"

7. P. 5, Line 5 "substantial mislocations" does not seem the proper word here, as this word implies some error in the location. However, you rather discuss here differences amomg centroid and hypocenters which may be true features. I would use the wording "substantial differences"

8. P. 5, Line 21 a rotation of 20% Kagan angle out of 120% maximal differences, and assuming this linearly maps into focal mechanisms similarity, should then be rather ~80% (not 90%)

9. Bottom of P. 5. While ISC reports both focal mechanisms and moment tensors, there is no discussion at all on the moment tensors, their similarity (the similarity is here discussed only using a Kagan angle, and thus concerns the double couple -DC terms onyl) and their non-DC terms, and all the discussion is done on DC components. I think this is fair, if you make this clearly explicit.

---

## Referee Comment (RC2) · Anonymous Referee #2 · 28 Feb 2019

The submitted manuscript, in this present form, gives general information about ISC bulletins, focal mechanism compilation. Such information is considered to be useful for users. However, some points which I listed below about the focal mechanisms selection are not clear.

I leave the final decision to the editor, but my decision is a major revision for the manuscript.

Comments:

1 - There are no major differences in the solutions of the focal mechanisms for large earthquakes in ISC bulletins. However, there are serious differences in small (M<5) and sometimes in moderate earthquakes. In this case, how will the user decide which

solution is correct for smaller earthquakes using ISC bulletins? This study has tried to find a solution to this kind of question. But the answer to the question is not given. Only a discussion was made in the sense of depth and rotation angle. But still, the main question is not answered. If ISC calculates focal mechanism using the first motion polarities for small earthquakes (M> 3.5), it will be more reliable as in earthquake locations given by ISC. I think it would be more appropriate to develop the study in this direction.

2 – The study tries to determine which focal mechanism should be used by the user in multiple mechanism solutions. However, the article content does not include such a conclusion.

3 - The figures cannot go beyond giving statistical information and them away from the main purpose of the manuscript.

4 – It is not clear in figure 5 how to choose the proper or target focal mechanism for earthquakes with multiple focal mechanism solutions. This is considered to be a significant shortcoming.
* * *

---

## Author Comment (AC1) · 3 Apr 2019

We have taken the Reviewers' comments into account in the preparation of this revised version of the paper, which contains some text changes and additions, to fully address their requests. Moreover, we extended the data time period covered in this manuscript until the end of the data year 2018, as opposed to October 2018 when the manuscript was first submitted. The data added until the end of 2018 does not change the content and results discussed. Here follows our response to the Reviewers' comments and an explanation of the points that we have changed. Attached to our response is the revised version of manuscript with changes tracked in blue for the text addition/changes and in red for the text deleted.

[Figure]

**1 Reviewer 1**

**1.1 "ISC computed source mechanisms"**

Reviewer 1 expressed the query on why the ISC computed focal mechanism solutions are not discussed in the paper even though they are briefly mentioned in the abstract. The Reviewer claimed that it does not make sense to omit a part of the source mechanism solutions that can be found in the ISC Bulletin since the scope of the paper is to discuss the ISC as a comprehensive source of information on earthquake source mechanisms.

We appreciated the Reviewer's comment and have now included the ISC computed solutions in our analysis throughout the paper. Text has been added in Section 2 (page 3, lines 3-9) and in Section 5 (page 8, lines 27-29) and all the figures in the revised manuscript have been rebuilt in order to include the ISC computed focal mechanism solutions. The additional data did not change the main points of this paper.

**1.2 "ISC and other global sources"**

Reviewer 1 suggested to present more information on the reported mechanism solutions by each agency in the ISC Bulletin in terms of geographical distribution, magnitude range, number of reported mechanisms, time distribution and provided information. The Reviewer proposed to do so at least for the agencies that report solutions more systematically, covering a long period of time in order to point out differences and complementarity.

Information regarding the number of reported solutions, type of methodology being applied (where available) and time distribution has already been presented in Figure 2. In order to address the rest features in the Reviewer's comment we added Table 1 where

we present the magnitude range covered by each agency and the type of parametric data that is being reported by each agency. Moreover, we added Figure 4 which shows the geographical and magnitude distributions covered by the most systematically reporting agencies hoping to help the users of the ISC Bulletin to identify areas where local/regional agencies offer complementary information to global agencies. Text is also added in Section 2 (page 3, lines 11-25) to give some examples in specific areas on how local agencies cover gaps in magnitude range left by global agencies. Details on the methodologies being applied by different agencies are also presented in Section 3 (page 4, lines 25-35, and page 5, lines 1-15). Implications of combining solutions determined by different agencies using different techniques are discussed in Section 5 (see for example page 8, lines 30-36, and page 9, lines 4-10).

**1.3 "Section 3 restructuring"**

Reviewer 1 had objections regarding the structure of Section 3 entitled "Source mechanism variability" and proposed to restructure it and further discuss some parts of it. Specifically, the Reviewer suggested to start by discussing the point source approximation as a concept, discuss the different types of point source models (focal mechanisms, moment tensors) and finally the data being used in different methodologies (polarities, amplitude ratios, waveform modelling) and differences among moment tensors and centroid moment tensors, and centroid locations and hypocentre locations.

Moreover, the Reviewer suggested to further discuss a couple of examples briefly mentioned in the originally submitted manuscript (page 4, lines 8-12 of the original manuscript) and suggest if possible how to select a reference source mechanism for earthquakes with multiple solutions. Then, to discuss further the reasons why some earthquakes show inconsistency between the reported mechanisms and finally, to give more information on how the fault styles in Figure 5 (in the original manuscript, Figure 6 in the revised manuscript) are obtained.
Following the Reviewer's comments, we have rewritten (almost entirely) Section 3 in order to comply with the suggestions mentioned above. Specifically, we added two new paragraphs in the beginning of Section 3 (page 4, lines 1-18) to address the Reviewer's comments on the point source approximation and point source models. The text has been reformed in page 4 (lines 25-35) and page 5 (lines 1-15) in order to discuss differences in moment tensors and centroid moment tensors, and centroid and hypocentre locations and make it easier for the reader to follow.

Text has been added in page 6 (lines 3-16) in order to discuss more thoroughly two case studies of large variability in reported source models as suggested by the Reviewer. New text in page 6 (lines 23-35) addresses the Reviewer's comments on identifying cases with large intra-event variability in source models and potential reasons that may contribute in these cases (see also page 6, lines 19-22).

Text in Section 5 (page 8, lines 25-36, and page 9, lines 4-18) explains further the scope of this paper, discusses complexities in different source models and gives some general hints on data quality characteristics which can help user of the ISC Bulletin to select a source model that may be more robust or more appropriate for their research.

Finally, we expanded the text at page 7 (lines 1-25) to better explain how the annual counts of Figure 6 (Figure 5 in the original manuscript) are obtained following the fault styles of Zoback (1992). Examples to events in the ISC Bulletin have also been added with links to specific events.

**1.4  "Figures"**

Reviewer 1 suggested a few improvements in the existing figures, namely:

1. Figure 1: to show the temporal evolution of target magnitudes or and histograms discussing the magnitude distribution. - We split the data period covered in this

paper in six parts and added a subplot for each time period (decade) in order to show the time evolution in magnitude range coverage in the entire ISC Bulletin.

2. Figure 2: institute acronyms should be listed in an additional table. - We added Table 1 to address this comment.

3. Figure 3: add titles for the upper and bottom panels including information regarding agencies and minimum magnitude. - We addressed this comment accordingly.

4. Figure 5: add a symbol legend. - We added a legend in this figure (Figure 6 in the revised manuscript).

5. Figure 5: what are undefined earthquakes? - We explain this in the revised text (page 7, line 6)

**1.5 Other minor comments**

We addressed all the other points raised by Reviewer 1 as follows (following the same order as in the Reviewer's text):

1. page 1, line 8: clarify the meaning of 90% or reformulate the text. - We addressed this comment by adding more information (page 1, lines 8-9).

2. page4, line 5: maximum intra-event rotation (it needs better explanation, introduce the concept clearly). - We added new text in page 5 (lines 28-31) to address this comment.

3. page 4, line 14: "...can be as high as $100°$". - We added new text in some earlier stage (page 5, lines 24-28) to address this comment and give an a physical explanation for the rotation angle.

[Figure]

4. page 5, line 8: "be aware of the techniques being used." Is the bulletin reporting details in techniques, data, azimuthal coverage? - The ISC Bulletin reports the number of stations and errors in the collected source models where available. This varies from agency to agency. This is partly a reason for the existence of this paper, thus, to make users aware of this information. Text was added in pages 6 (lines 20-22) and 8 (lines 24-29).

5. page 5, line 12: "centroid or not" should be "centroid or hypocentre". - We corrected this (page 8, line 22).

6. page 5, line 10: "substantial mislocations" should be "substantial differences". - We corrected this (page 8, line 19).

7. page 5, line 21: should be 80% (not 90%). - This is right, we corrected it (page 9, line 23).

8. bottom of page 6: the similarity of both focal mechanisms and moment tensors is discussed only using a Kagan angle and the discussion is done on DC components. - We added some text to clarify this (page 9, lines 24-27).

**2   Additional changes**

We added a few new references to the manuscript, updated the figures and corresponding captions and at times streamlined the text.

Please also note the supplement to this comment:
https://www.earth-syst-sci-data-discuss.net/essd-2018-143/essd-2018-143-AC1-supplement.pdf

---

## Author Comment (AC2) · 3 Apr 2019

We have taken the Reviewer's comments into account in the preparation of this revised version of the paper, which contains some text changes and additions, to fully address their requests. Moreover, we extended the data time period covered in this manuscript until the end of the data year 2018, as opposed to October 2018 when the manuscript was first submitted. The data added until the end of 2018 does not change the content and results discussed. Here follows our response to the Reviewer's comments and an explanation of the points that we have changed. Attached to our response is the revised version of manuscript with changes tracked in blue for the text addition/changes and in red for the text deleted.

[Figure]

**1 Reviewer 2**

**1.1 "Which mechanism to use?"**

Reviewer 2 mentions that the manuscript does not offer an answer in the question which mechanism to use in the case of multiple solutions for the same earthquake. In fact this question is mentioned in two out of four comments. The Reviewer claims that the current paper has tried to answer this question but the answer is not given. The Reviewer also proposed to develop the manuscript towards the direction of presenting the ISC computed focal mechanisms as more reliable, similar to the ISC event locations.

Let us first clarify that the purpose of this manuscript is an attempt to draw the Researchers' attention on the availability of source models in the ISC Bulletin. It is true that for decades the ISC put all of the effort in analysing and relocating seismic events with the use of reported parametric data, whilst not much, if any, has been done to promote the availability of collected source models. In fact the usage of the dedicated online web search on source mechanisms in 2018 corresponds to less than 2% of the total use of the ISC web site!

In this paper we aim to acknowledge the agencies reporting source models in the ISC Bulletin, discuss the differences in techniques being used by the major contributors, and highlight the complexities in combining source models following different concepts and being determined by different techniques. For this reason we added new text in Section 3 (page 4, lines 1-18) and pages 8 (lines 24-36) and 9 (lines 4-19), also combining some of the comments posed by Reviewer 1. Moreover, in order to highlight how different agencies can act complementary to each other and provide a more comprehensive coverage in space, time and magnitude we added Figure 4 and new text in page 3 (lines 11-25).

Regarding the ISC computed focal mechanisms, we added these mechanism solutions

in the analysis of the revised manuscript (also suggested by Reviewer 1) but we cannot claim that the ISC focal mechanism solutions are more reliable than other solutions provided by other agencies, since they can be fundamentally different in comparison with them. Take as an example a centroid based moment tensor solution (see for example additional text in page 4, lines 1-18, and page 6, lines 3-22). Finally, given all the above explained complexities, we gave only general recommendations described in page 8 (lines 24-36) and page 9 (lines 4-19). Yet again, we have made clear in the manuscript that it is not our purpose to recommend the use of one data provider over another, but we provide all the information we can collect and let the Researchers to choose this with respect to the needs of their research.

**1.2 "Figures"**

Reviewer 2 commented that the Figures only cover statistical information and do not serve the purpose of the manuscript. We believe that enhancing the manuscript with new text, it is clearer and more obvious how the Figures presented in this paper can help readers to better understand the content of the ISC Bulletin regarding the availability of earthquake source models. The newly added Figure 4 should also help further towards this direction.

**1.3 "Figure 5"**

In conjunction with Reviewer's 2 first comment, more information was requested on how to choose a proper or target mechanism solution in Figure 5 for earthquakes with multiple solutions.

To address this point we have expanded the text first by recalling the fault styles according to Zoback (1992) and secondly by summarizing how we proceeded to characterize an event when multiple solutions are available. Examples for different cases have been

given and links to real cases in the ISC Bulletin added. We believe that the revised text clarifies the scope of original Figure 5 (now Figure 6 in the revised manuscript). The revised text regards the entire page 7.

**2 Additional changes**

We added a few new references to the manuscript, updated the figures and corresponding captions and at times streamlined the text.

Please also note the supplement to this comment:
https://www.earth-syst-sci-data-discuss.net/essd-2018-143/essd-2018-143-AC2-supplement.pdf

[Figure]

**Supplement:**

[revised manuscript text omitted]